DATA RELEASE

# Linking gut microbiome to bone mineral density: a shotgun metagenomic dataset from 361 elderly women

Qi Wang[1,2,†], Qiang Sun[1,3,†], Xiaoping Li[1], Zhefeng Wang[4], Haotian Zheng[1,2], Yanmei Ju[1,2], Ruijin Guo[1,5], Songlin Peng[4,*] and Huijue Jia[1,6,*]

1 BGI-Shenzhen, Shenzhen 518083, China
2 School of Future Technology, University of Chinese Academy of Sciences, Beijing 101408, China
3 Department of Statistical Sciences, University of Toronto, Toronto, Canada
4 Department of Spine Surgery, Shenzhen People's Hospital, Ji Nan University Second College of Medicine, 518020 Shenzhen, China
5 Macau University of Science and Technology, Taipa, Macau 999078, China
6 Shenzhen Key Laboratory of Human Commensal Microorganisms and Health Research, BGI-Shenzhen, Shenzhen 518083, China

**Submitted:** 01 June 2020

* Corresponding authors. E-mail: psling824@163.com; jiahuijue@genomics.cn

† Contributed equally.

Preprint submitted at https://doi.org/10.1101/679985

## ABSTRACT

Bone mass loss contributes to the risk of bone fracture in the elderly. Many factors including age, obesity, estrogen and diet, are associated with bone mass loss. Mice studies suggested that the gut microbiome might affect the bone mass by regulating the immune system. However, there has been little evidence from human studies. Bone loss increases after menopause. Therefore, we have recruited 361 Chinese post-menopausal women to collect their fecal samples and metadata to conduct a metagenome-wide association study (MWAS) to investigate the influence of the gut microbiome on bone health. Gut microbiome sequencing data were produced using the BGISEQ-500 sequencer. Bone mineral density (BMD) was calculated using a Hologic dual energy X-ray machine, and body mass index (BMI) and age were also recorded. This collected data allows exploration of the gut microbial diversity and their links to bone mass loss as well as to microbial markers for bone mineral density. In addition, these data are potentially useful in studying the role that the gut microbiota might play in bone mass loss and in exploring the process of bone mass loss.

**Subjects** Genetics and Genomics, Metagenomics, Medical Microbiology, Molecular Infection Biology

## CONTEXT

Bone mass loss is a process where calcium and phosphate from the bones is reabsorbed into the rest of the body instead of being retained in the bones, making our them weaker [1]. It is a severe and common condition among the elderly, especially during menopause, when estrogen loss occurs and bone mass decreases, thus increasing the risk of bone fracture, which can result in acute pain and even death [2]. Recently a new concept, "osteoimmunology", has revealed tight interactions between the immune system and bone metabolism [3]. Interestingly, it has been widely recognized that the gut microbiota could influence host health by interacting with the host immune system [4]. But most related research [4, 5] has been carried out using mice and 16S sequencing, which is of poor taxonomic resolution, low sensitivity, and contains no functional related information [6].

Qi Wang[1,2], Qiang Sun[1,3], Xiaoping Li[1], Zhefeng Wang[4], Haotian Zheng[1,2], Yanmei Ju[1,2], Ruijin Guo[1,5], Songlin Peng[4], Huijue Jia[1,6]

[1]BGI-Shenzhen, Shenzhen 518083, China;
[2]School of Future Technology, University of Chinese Academy of Sciences, Beijing, 101408, China.;
[3]Department of Statistical Sciences, University of Toronto, Toronto, Canada;
[4]Department of Spine Surgery, Shenzhen People's Hospital, Ji Nan University Second College of Medicine, 518020, Shenzhen, China.;
[5]Macau University of Science and Technology, Taipa, Macau 999078, China;
[6]Shenzhen Key Laboratory of Human Commensal Microorganisms and Health Research, BGI-Shenzhen, Shenzhen 518083, China

**Figure 1.** Protocol collection for sequencing and analysing female bone mass loss from the microbiota [7]. https://www.protocols.io/widgets/doi?uri=dx.doi.org/10.17504/protocols.io.bq9kmz4w

Shotgun sequencing is of high enough resolution to explore the relationship between the gut microbiome and bone loss at both the species and functional levels. In our study, we enrolled 361 postmenopausal women who did not use antibiotics in the one month prior to the study and came from different communities in Shenzhen. Their stool samples and multiple metadata related to bone loss were collected for a metagenome-wide association study (MWAS).

## METHODS

A protocol collection including methods for DNA sequencing, QC, and bioinformatics is available via protocols.io (Figure 1) [7].

## Sampling strategy

Samples were collected from May 20 to September 17, 2017 at the medical center of Shenzhen People's Hospital (Shenzhen, China). According to records, none of the volunteers used antibiotics within one month prior to the study. Fecal samples from each of the 361 post-menopausal women were collected and immediately frozen at −80 °C for storage (sTab 1 [8]). The samples were then transported on dry ice to BGI-Shenzhen, total DNA extraction, detection [9], and sequencing using the BGISEQ platform (BGISEQ, RRID:SCR_017979), which were conducted according to previously published protocols [10, 11]. The raw reads that had 50% low-quality bases (quality ≤ 20) or more than five ambiguous bases were excluded. The remaining reads were mapped to the human genome (hg19) by SOAP v2.22 (-m 100 -x 600 -v 7 -p 6 -l 30 -r 1 -M 4 -c 0.95) to remove the human DNA (sTab 3 [8]). The high-quality non-human reads were defined as cleaned reads following previous methods [12, 13].

## Ethics approval and consent to participate

This study was approved by the Institutional Review Board on Bioethics and Biosafety at Shenzhen People's Hospital (LL-KY-2019506) and BGI (BGI-IRB 19126). In addition, all the volunteers were fully informed of the significance and scientific value of the project, and they voluntarily agreed to sign informed consent forms for scientific use of the metadata.

Moreover, the informed consent included using information on phenotype. The sample names were also anonymized.

## Taxonomic and functional abundance calculation

The cleaned data were used for the annotation and profile acquisition of taxons using MetaPhlan2 (MetaPhlAn, v2.0, RRID:SCR_004915, code: metaphlan2.py input.fastq –input_type fastq –nproc 10 > profiled_metagenome.txt) [14]. We removed the species present in less than 10% of the samples for later analysis. For functional abundance calculation, the putative amino acid sequences were first translated from the gene catalogues [15] and aligned against the proteins/domains in the KEGG databases (release 79.0, with animal and plant genes removed) using BLASTP (v2.2.26, code: blastall -p balstp -m 8 -e 0.01 -a 6 -b 100 -K 1 -F T -m 8 -d database -o output -i input). Each protein was assigned to the KO group by the highest scoring annotated hit(s) containing at least one HSP scoring >60 bits. The relative abundance profile of KOs was determined by summing the relative abundance of genes from each KO [14, 16]. The KO abundance was used as the input file to calculate the profile of gut metabolic modules (GMMs, evaluating gut metabolic potential and anaerobic fermentation capacity). The code is shown as below: java -jar GMM/omixer-rpm-1.1.jar -d gmm_input.txt -i $OUTPUT_DIR/normalization.pr -s average -o $OUTPUT_DIR/ [17].

## Two-stage least squares regression analysis [18]

Stage 1: In the first step, we fit the relationship between the taxonomic abundance or metabolic module abundance and the age and BMI to a linear regression and saved the prediction value. Details of the taxonomic abundance are in the "Taxonomic abundance calculation" section (sTab 3 [8]). Details of the metabolic module abundance are in the "Gut metabolic modules analysis" section (sTab 4 [8]). This step was used to adjust for the effects of age and BMI to the contribution of BMD by taxonomic abundance or metabolic module abundance.

Stage 2: Five-fold cross-validation was performed ten times on a random forest regression model (Y: the BMD T score; X: the prediction value from the stage 1). The error curves from ten trials of five-fold cross-validation were averaged. We chose the model that minimized the sum of the test error and its standard deviation in the averaged curve.

## Alpha-diversity and count

The within-sample diversity was calculated via the Shannon index (sTab 4 [8]), as described previously [6]. A genes was considered present if more than one read mapped to it.

## DATA VALIDATION AND QUALITY CONTROL

The metagenomic shotgun sequencing of 361 samples was performed, obtaining an average of 7.7 gigabases (Gb) clean data per sample (sTab 3 [8]). To explore the utility of this data, the life and clinical index (sTab 1–2 [8]) to the T-score in our cohort was assessed and significant factors such as age and body mass index (BMI) to the microbiome were excluded, and the alterations of the gut microbiome along with the T-score were evaluated. Finally, a stable regression model was built at the species and module levels for the cohort. The T-score of the BMD in the lumbar spine was used to represent the bone mass.

## STATISTICAL ANALYSES

Generally, statistical significance was set to 0.05 and only patients with complete data were analyzed.

## REPRODUCIBLE RESEARCH

Others could reproduce the reported analysis from clean reads by the declared software and parameters [7, 8].

## RESULTS

The BMD was calculated from data obtained using a Hologic dual energy X-ray machine at Shenzhen people's Hospital (sTab 1 [8]). We used the T-score of BMD in the lumbar spine to represent the bone mass [19]. A sample's T-score is a relative measure of the sample's BMD compared to the reference population of young, healthy individuals with the same gender. The T-score is classified as normal (T-score of −1 or above), low (T-score of −1 to −2.5) or osteoporosis (T-score < −2.5).

### Result 1. A mild gut microbiome dysbiosis has been seen for bone mass loss

To explore alterations of the gut microbiome along with the change in the T-score, the change in different taxonomy levels was analyzed. Diversity at gene ($p = 4.53 \times 10^{-9}$, adjusted $R^2 = 0.0904$, linear regression, Figure 2b, sTab 4 [8]), species ($p = 1.17 \times 10^{-15}$, adjusted $R^2 = 0.162$, linear regression, Figure 2d, sTab 4 [8]), and genus ($p = 7.98 \times 10^{-14}$, adjusted $R^2 = 0.144$, linear regression, Figure 3b, sTab 4 [8]) levels increased with the T-score and was probably caused by an increase of pathogenic microcells in the gut. In addition, the count data also showed an increase in the T-score at the gene ($p = 0.0114$, adjusted $R^2 = 0.0152$, linear regression, sFigure 2a), species ($p = 2.33 \times 10^{-5}$, adjusted $R^2 = 0.0465$, linear regression, Figure 2c, sTab 5 [8]), and genus ($p = 7.73 \times 10^{-10}$, adjusted $R^2 = 0.0992$, linear regression, Figure 3a, sTab 6 [8]) levels. To further characterize the changes, the top 20 most abundant species were chosen (Figure 1c). The data demonstrated that the *B. stercoris, E. coli, B. uniformis, B. coprocola, B. fragilis , E. rectale,* and *E. eligens* were significantly negatively associated with the T-score, while data for *B. vulgatus, B. massiliensis, B. caccae* and *Megamonas unclassified* displayed an obvious positive correlation with the T-score (Figure 3c). In addition, in the top 15 abundant genera, the *Eubacterium, Escherichia, Subdoligranulum, Klebsiella, Clostridium* and *Blautia* were significantly negatively correlated with the T-score (Figure 4, sTab 6 [8]). Among these genera, *Eubacterium* and *Escherichia* are normal microorganisms of the gut and can cause infection under opportunistic conditions. Positively correlated genera included the *Prevotella, Parabacteroides, Megamonas* and *Akkermansia* (Figure 3). For the top 10 enriched phyla, the *Bacteroidetes, Verrucomicrobia, Fusobacteria, Euryarchaeota* and *Ascomycota* were positively correlated with the BMD T-score (Figure 5, sTab 7 [8]), while the *Proteobaccteria, Actinobacteria, Synergistetes* and *Chlorobi* were negatively correlated (Figure 5).

### Result 2. Species linked to BMD

To select the species with strong correlations using the T-scores, we used the two-stage least square method [18] to regress the species to the T-score (details are shown in the Methods section). The model showed a high R-squared value (>0.99, sTab 3a [8], Figure 6a), and 18

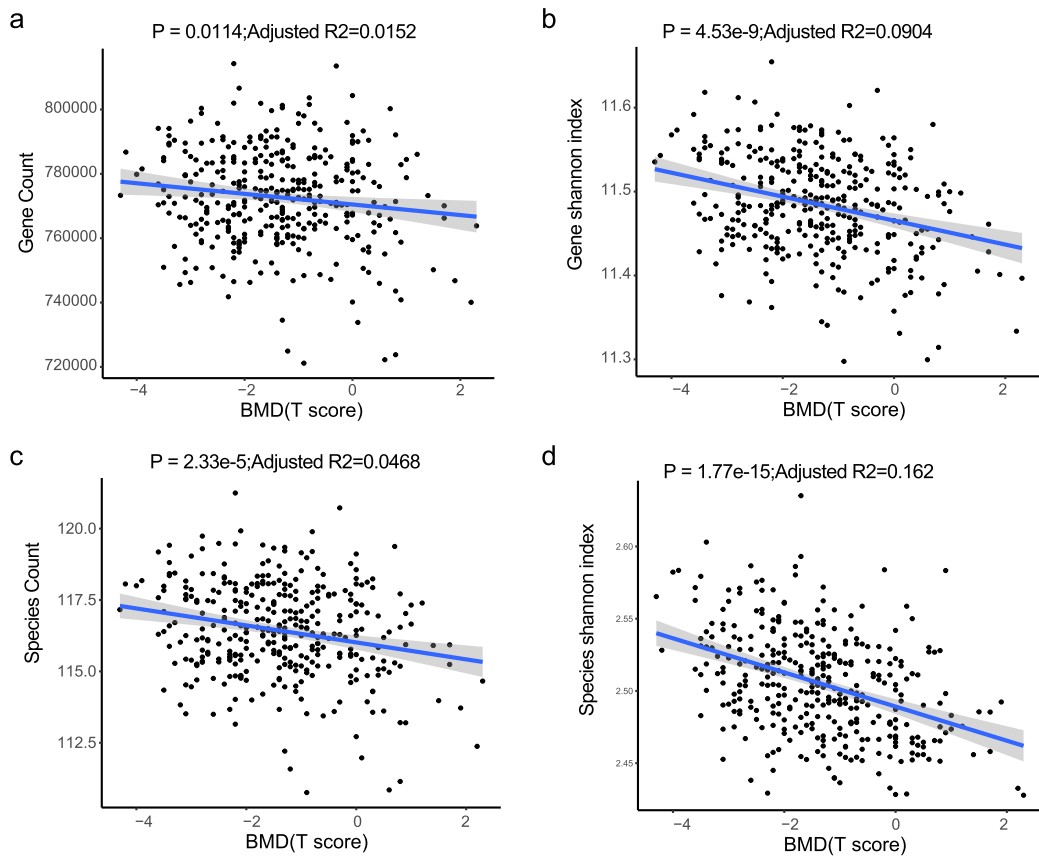

**Figure 2.** Slight increase in gut microbial diversity. (a,b) Count and Shannon index at the gene and species level of the two cohorts (liner regression).

species were selected. These species were ranked by their importance (Figure 6b). The Spearman's rank correlation was used to evaluate the relationship between the selected species and the clinical indexes (Figure 6c). From these findings, it was easy to see that some T-score negatively correlated species, including *Streptococcus parasanguinis, Clostridium perfringens, Haemophilus sputorum, Enterobacter aerogenes, Actinobacillus unclassified,* and *Chlorobium phaeobacteroides* were negatively connected with triglyceride (TG) levels, but were positively correlated with β-Crosslaps (CROSSL) and high-density lipoprotein (HDL). Meanwhile, some T-score positively correlated species, such as *Roseburia intestinalis,* a butyrate-producing bacterium, could influence the human immune system [20]. *Enterobacter cloacae* and *Sutterella wadsworthensis* were positively correlated to TG but were negatively to CROSSL and HDL.

## Result 3. Functional modules indicating bone mass loss

To find the functional modules highly correlated with the T-score, the two-stage least square method was used [18]. Thirteen modules with the R-squared values higher than 0.99 (sTab 3a [8], Figure 7a) were obtained by the model and plotted in rank by their importance (Figure 7). In addition, the negatively correlated functional modules, like lactate consumption, sucrose degradation, and tryptophan degradation were positively associated



**a**

*P* = 7.73e-10;Adjusted R2=0.0992

Genus Count vs BMD(T score)

**b**

*P* = 7.98e-14;Adjusted R2=0.144

Genus Shannon index vs BMD(T score)

**c**

Spearman` correlation
0.4
0.2
0
−0.2
−0.4

Megamonas unclassified *
Eubacterium eligens *
Eubacterium rectale *
Bacteroides caccae *
Klebsiella pneumoniae *
Bacteroides fragilis *
Subdoligranulum unclassified *
Bacteroides coprocola *
Bacteroides dorei *
Bacteroides thetaiotaomicron
Bacteroides massiliensis *
Alistipes putredinis +
Bacteroides uniformis *
Bacteroides ovatus
Escherichia coli *
Bacteroides plebeius
Bacteroides vulgatus *
Bacteroides stercoris *
Prevotella copri
Faecalibacterium prausnitzii

BMD(T score)

The adjusted relative abundance(%)

**Figure 3.** Slight increase in gut microbial richness. (a,b) Richness and alpha-diversity (Shannon index) at the genus level of the two cohorts (liner regression). (c) The top 15 species. (The Spearman's correlation, "+" for *p* < 0.05; "*" for *p* < 0.01).

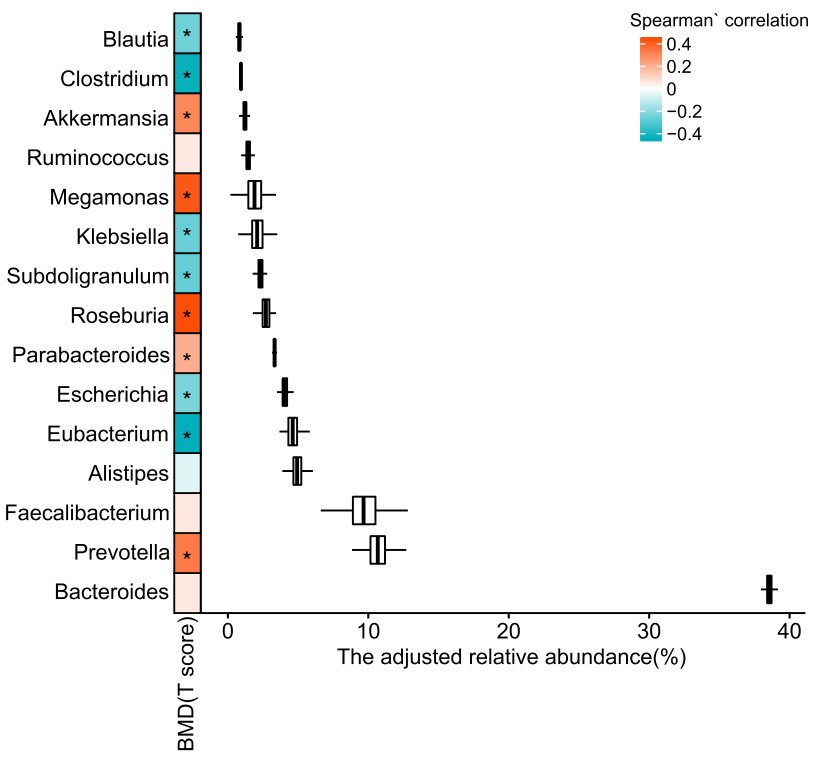

**Figure 4.** Means of the top 15 most abundant genera. The Spearman's correlation, "*" for *p* < 0.01.

with HDL and CROSSL, but negatively correlated with TG. In contrast, the BMD positively correlated functional modules, such as pectin degradation, trehalose degradation, arginine degradation (which can prevent bone mass loss and bone collagen breakdown in rat model [21]), mucin degradation, and rhamnose degradation were positively associated with TG, but negatively correlated with HDL and CROSSL. More detailed information on these functional modules can be found in Figure 8.

## DISCUSSION

We carried out the first study to explore the alteration of the gut microbiome along with bone mass loss in 361 post-menopause Chinese urban women with MWAS. Firstly, taxonomy diversity was observed to increase at various levels, which could have resulted from growth of some opportunistic pathogens in the gut. In addition, some T-score highly correlated species and functional modules were also revealed by our study, which might offer new strategies for better diagnosis and mechanistic understanding of bone mass loss.

For the volunteers, we tried to exclude the influence of gender, region, and antibiotics on the bacterial flora, as post-menopausal women in Shenzhen who had no history of antibiotic use within the one month before the study were selected. Furthermore, the sample size was large enough for us to resolve the change in the gut microbiome along with bone mass loss. Our data suggest that the gut microbiome is closely related to the process of bone mass loss in post-menopausal urban women in China. Although the mechanism of how the gut microbes affect and modulate bone metabolism is not fully understood, our

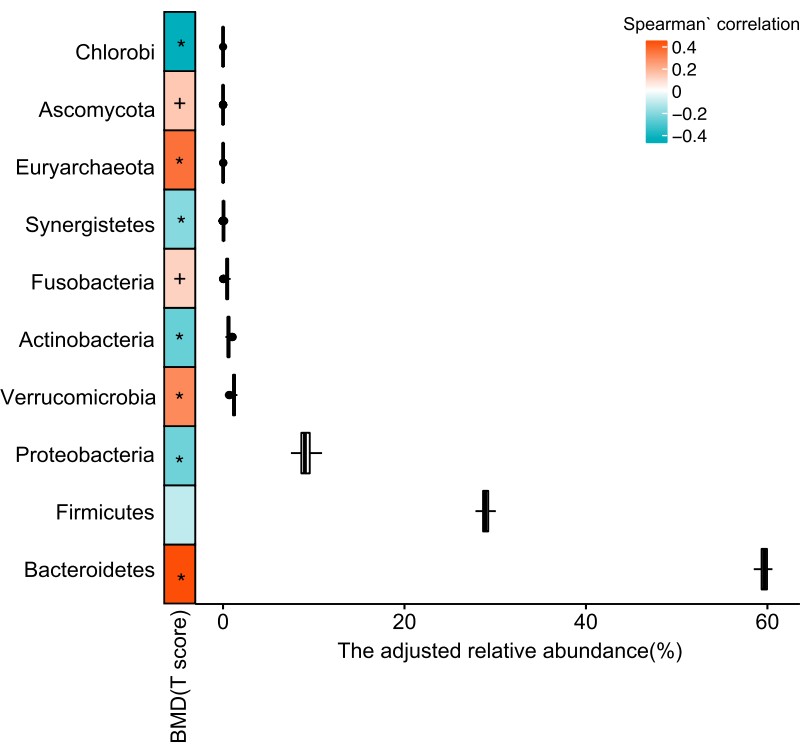

**Figure 5.** Means of the top 10 most abundant genera. The Spearman's correlation, "+" for $p < 0.05$; "*" for $p < 0.01$.

research indicates that gut microbiota may be novel targets for the prevention of bone mass loss and provide a new avenue for future studies and treatment in this field.

## POTENTIAL RE-USE

This is the first dataset where 361 high-quality metagenomics datasets were collected from elderly Southern Chinese urban women. All the clinical indexes have also been provided in the table in GigaDB [8]. By initial exploration of these data, we can see a slight correlation between the gut microbiome and the bone mass loss. Meanwhile, we have also found some biomarkers related to the bone mass loss at both the species and function levels. For potential re-use, the clinical details collected here, such as the relationship between tea-drinking and bone mass or metagenomics, make this dataset valuable for further analysis. While we were unable to clearly determine any strong signals between the gut microbiome and bone mass loss with our methods, we hope that others can find novel insight in this dataset by using different statistical approaches. With the moderate sample size and detailed information on a number of clinical features, it might be a useful dataset to combine with and/or compare to other gut microbiome datasets.

## DATA AVAILABILITY

The filtered non-human DNA reads have been deposited at EBI (bioproject number PRJNA530339) and the CNGB CNSA [22] database (accession code CNP0000398). Abundance and other tabular data and a STORMS (Strengthening The Organizing and Reporting of Microbiome Studies) checklist is available in the *GigaScience* GigaDB repository [8].

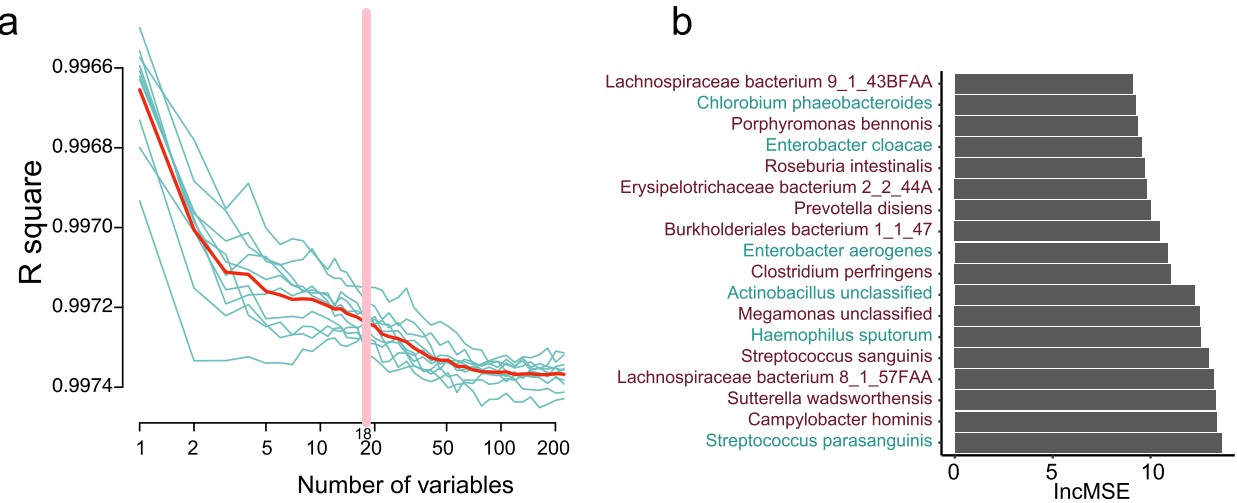

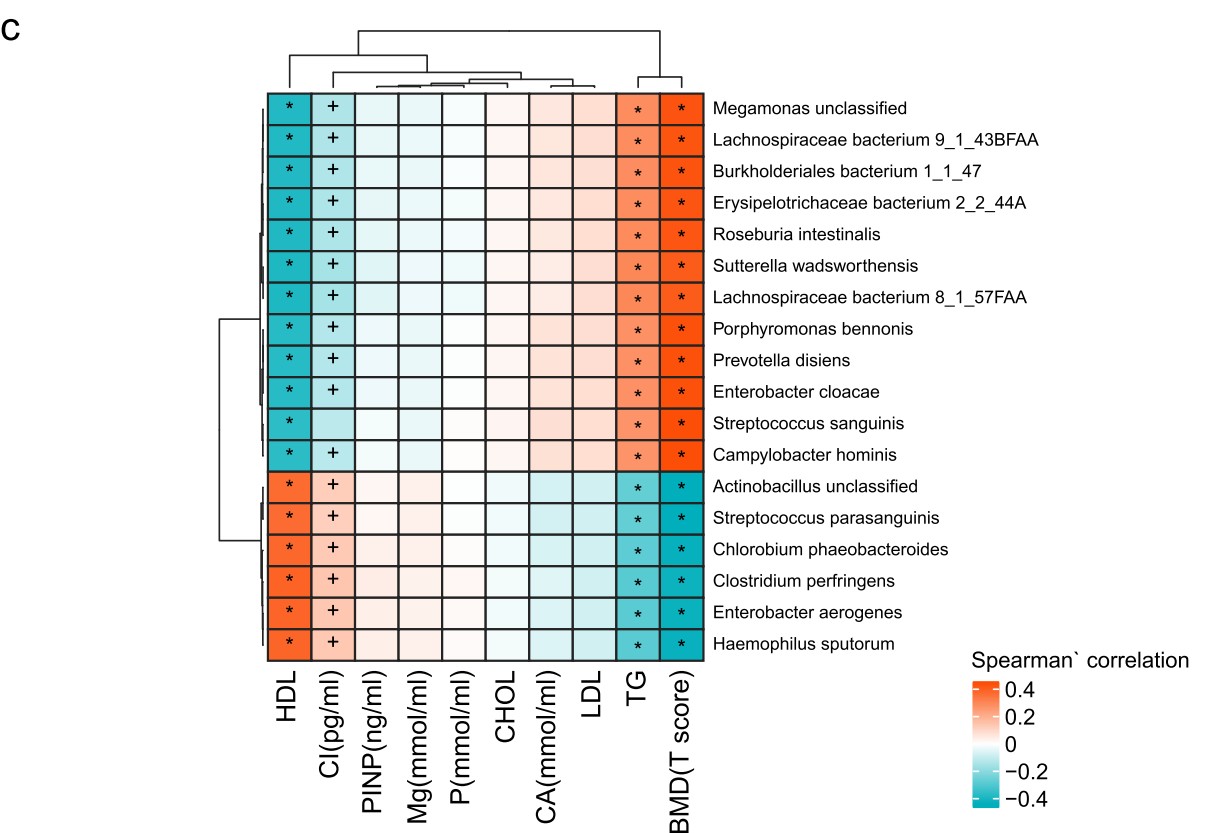

**Figure 6.** Fecal microbial species markers for BMD. (a) The R-squared value during the ten-time cross-validation process (the blue lines show the ten different process, the red line shows the average of the ten-time cross validation, and the pink line shows the best variables). (b) The lncMSE of the 18 chosen species markers. (c) The correlation between the markers and the clinical indeces. (Spearman' correlation, "+" for $p < 0.05$; "*" for $p < 0.01$).

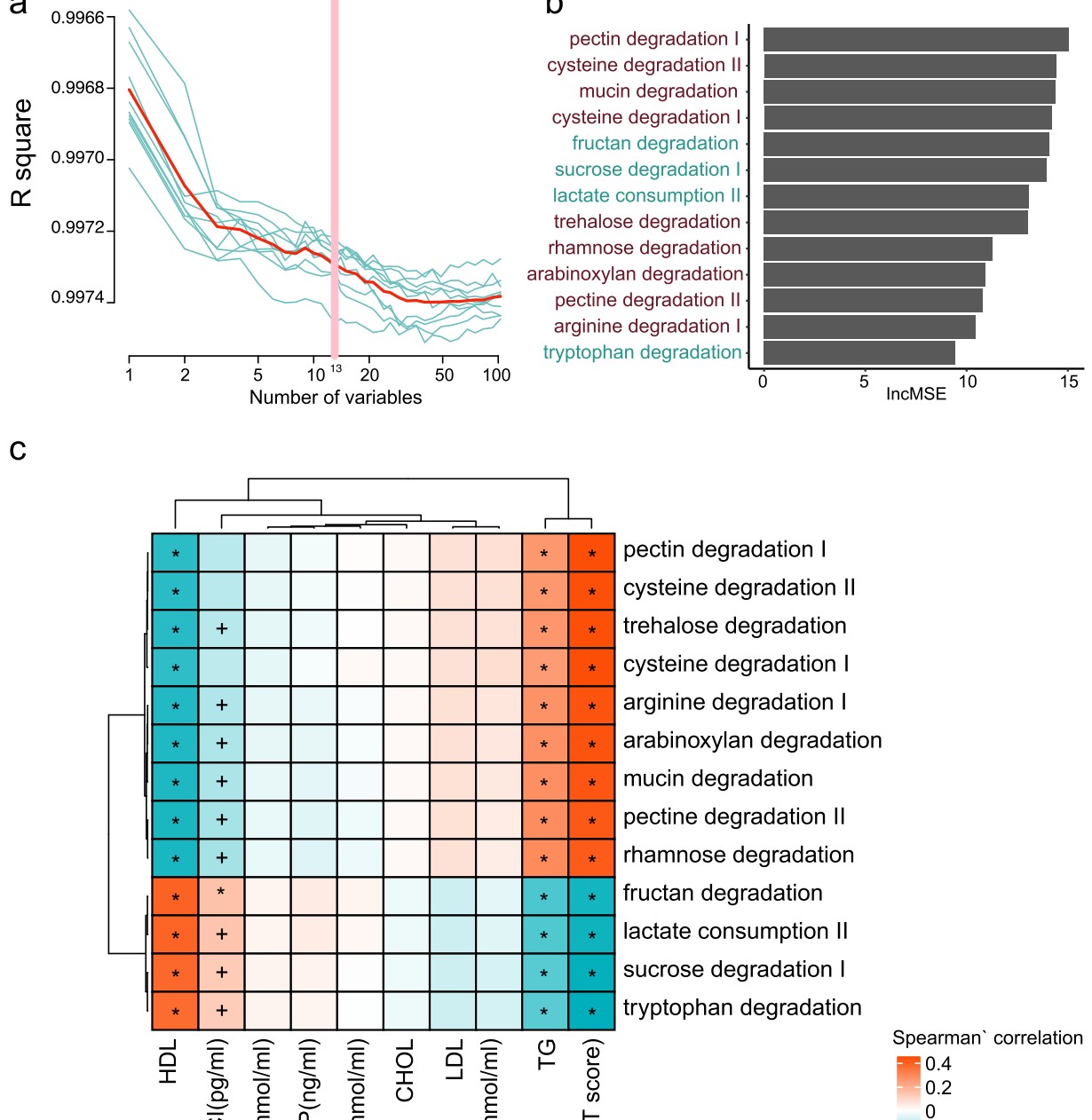

**Figure 7.** Fecal microbial modules markers for BMD. (a) The *R*-squared values during the ten-time cross-validation process (the blue lines show the ten different process, the red line shows the average of the ten-time cross validation, and the pink line shows the best variables). (b) The lncMSE of the 13 chosen modules markers. (c) The correlation between the markers and the clinical indeces. (Spearman' correlation, "+" for *p* < 0.05; "*" for *p* < 0.01).



**Figure 8.** The heatmap of gut metabolic modules with clinical indeces. Spearman' correlation, "+" for *p* < 0.05; "*" for *p* < 0.01.

## COMPETING INTERESTS

Qi Wang, Qiang Sun, Xiaoping Li, Haotian Zheng, Yanmei Ju, Ruijin Guo, Huijue Jia are employees of BGI.

## FUNDING

This work was financially supported by the Science, Technology and Innovation Commission of Shenzhen Municipality under grant No. JCYJ20160229172757249 and No. JCYJ20170817145523036.

## ABBREVIATIONS

BMD: Bone Mineral Density; BMI: Body Mass Index; CROSSL: β-crosslaps; Gb: Gigabase; GMM: Gut Metabolic Module; HDL: High-density Lipoprotein; MWAS: Metagenomic-wide Association Study; TG: Triglyceride.

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
