## [Reviewer Report]

Upload additional filesDRR-20201005/form/2020.9.15.for.gigabyte_review.docxReviewer name and names of any other individual's who aided in reviewer Levi WaldronDo you understand and agree to our policy of having open and named reviews, and having your review included with the published papers. (If no, please inform the editor that you cannot review this manuscript.)YesIs the language of sufficient quality?YesPlease add additional comments on language quality to clarify if needed
Are all data available and do they match the descriptions in the paper? NoAdditional CommentsThe checklist states “no raw reads for ethical” but the manuscript states “The sequencing reads from each sequencing library have been deposited at EBI with the accession number: PRJNA530339 and the China National Genebank (CNGB), accession number CNP0000398.” so there is a disconnect. Assuming human sequence reads are removed from the data, I’m not convinced of ethical reasons not to post microbial sequence reads, but it seems the authors have posted the microbial sequence reads.Are the data and metadata consistent with relevant minimum information or reporting standards? See GigaDB checklists for examples <a href="http://gigadb.org/site/guide" target="_blank">http://gigadb.org/site/guide</a>YesAdditional CommentsIs the data acquisition clear, complete and methodologically sound?YesAdditional CommentsIs there sufficient detail in the methods and data-processing steps to allow reproduction?YesAdditional CommentsIs there sufficient data validation and statistical analyses of data quality? YesAdditional CommentsIs the validation suitable for this type of data?YesAdditional CommentsIs there sufficient information for others to reuse this dataset or integrate it with other data?NoAdditional CommentsI appreciate that data have been posted at EBI and CNGB. Could the authors also comment on whether the metadata essential to the analysis are also provided, and that these can be linked to the sequence data? Although I’m glad to hear that “Others could reproduce the reported analysis from clean reads by the declared software and parameters” I do think that the code to reproduce the analysis should also be reported. Any Additional Overall Comments to the AuthorSee uploaded peer review.RecommendationMinor Revision

---

## [Reviewer Report]

Reviewer name and names of any other individual's who aided in reviewer Chris HunterDo you understand and agree to our policy of having open and named reviews, and having your review included with the published papers. (If no, please inform the editor that you cannot review this manuscript.)YesIs the language of sufficient quality?YesPlease add additional comments on language quality to clarify if needed
Are all data available and do they match the descriptions in the paper? NoAdditional Commentsmost of the data are provided as supplemental files in biorXiv, but in Excel rather than CSV. These data files will need to be curated into a GigaDB dataset.Are the data and metadata consistent with relevant minimum information or reporting standards? See GigaDB checklists for examples <a href="http://gigadb.org/site/guide" target="_blank">http://gigadb.org/site/guide</a>YesAdditional CommentsIs the data acquisition clear, complete and methodologically sound?NoAdditional Commentsthe consent by the patients to openly share all metadata is not clearly stated, simply saying the study was approved by the bioethics review board does not mean consent was given to share the data, just that the institute consent to the study being done.Is there sufficient detail in the methods and data-processing steps to allow reproduction?NoAdditional CommentsMaybe to someone with a good understanding of statistics there is sufficient detail, this is an area that a statistician should look at. 
For me, the descriptions of the analysis and the methods do not given anywhere near enough detail for me to either understand what was done or replicate it.
The concept of "Gut metabolic modules" is not defined here, with just a reference to another paper, a brief explanation of what is meant by the term here would be useful.Is there sufficient data validation and statistical analyses of data quality? YesAdditional Commentsthe sequences were filtered for human contaminants and adapter seq, also low quality reads were removed.Is the validation suitable for this type of data?YesAdditional CommentsIs there sufficient information for others to reuse this dataset or integrate it with other data?NoAdditional CommentsThe metadata is extensive but there are some basic points that are missing; collection date, antibiotic use, relatedness of samples/patients. Other less important details are also missing, like why and how this cohort was selected.Any Additional Overall Comments to the Author- I am concerned about the open sharing of patient metadata without the evidence that it was consented prior to sharing. 
- A lot of metadata is collected and provided in the supplemental tables (which is great for reuse) but there are no explanations of what the values are, while some headers are self explanatory others less-so e.g. what is CROSSL(pg/ml)? or "Side crops", 
- how were the various conditions diagnosed? 
- I see no indication of antibiotic usage in the cohort
- Are all the samples from different individuals? was each sample a single bowl movement?
- There is no background given as to how this cohort was selected or why.
- The is no discussion of the bone mass density of a "normal" cohort, does this cohort represent a normal cohort or is it already biased toward low or high density? Simply describing the cohort with respect to Normal (T of -1 or above), low (-1to-2.5) or osteoporosis (< -2.5) would be a help. I cannot see the T-scores included in the sTab1a file, are they computed from the L1-L4(z) values given?
- There are a number of NA values in the table of samples metadata, but there is no explanation as to how these samples where handled in the analysis.
- In general I feel that there is a lot of poorly described statistical analyses included that are not required as part of a data note, the focus should be on describing the data and ensuring the data and metadata are well explained.RecommendationMajor Revision